# SUBGRAPH PLUG-IN BOOSTS UP GRAPH NEURAL NETWORKS

## ABSTRACT

Message-passing neural networks (MPNNs) often collapse into a one-dimensional subspace because repeated neighborhood aggregation amplifies the dominant eigenvector of the normalized adjacency matrix, erasing local distinctions and limiting graph classification performance. In this paper, we theoretically analyze this phenomenon using perturbation theory to trace the eigenvector amplification process and mutual information bounds to quantify the resulting loss of discriminative signals. Guided by these insights, we propose the Subgraph Plug-in (SP), a lightweight, architecture-agnostic module that selects the top-$\kappa$ nodes by centrality, extracts their $\tau$-hop neighborhoods as interpretable subgraphs, and concatenates the resulting subgraph embeddings with the global representation of any base GNN without altering its architecture or incurring significant computational overhead. Across 11 graph-classification benchmarks and 13 GNN variants, we evaluate each backbone with and without SP, yielding 110 model–dataset pairs; SP improves performance in 94 of 110. Beyond these, on ZINC and OGBG-MolHIV, we conduct head-to-head comparisons against 11 methods, including augmentation modules, recent GNNs, and subgraph-based methods. SP achieves the best results among augmentation and subgraph-based approaches and remains competitive with recent GNNs, supporting its role as a widely applicable, cost-effective plug-in that preserves feature diversity and amplifies discriminative substructures. performance.

## 1 INTRODUCTION

Graph classification is essential for diverse domains, from drug discovery to traffic network analysis, where predicting global properties depends on graph topology Zhou et al. (2020); Wu et al. (2022). Modern approaches rely mainly on message-passing neural networks (MPNNs), which iteratively update node features via neighborhood aggregation:

$$H^{(l+1)} = \sigma\big(\tilde{A}\,H^{(l)}W^{(l)}\big). \tag{1}$$

Here, $\hat{A} = A + I$, $\hat{D}_{ii} = \sum_j \hat{A}_{ij}$, $\tilde{A} = \hat{D}^{-1/2}\hat{A}\hat{D}^{-1/2}$ is the normalized adjacency matrix, and $W^{(l)}$ denotes the learnable weights at the layer $l$ Kipf & Welling (2016).

Despite strong empirical results and the common practice of stacking more layers to expand the receptive field, deep MPNNs suffer from a fundamental collapse: as the number of layers $l$ grows, repeated propagation along the normalized adjacency $\tilde{A}$ amplifies its dominant eigenvector such that

$$\lim_{l \to \infty} \left\| \frac{H^{(l)}}{\|H^{(l)}\|_F} - Y \right\|_F = 0. \tag{2}$$

Here, $Y$ is rank-one (i.e., $\text{rank}(Y) = 1$ and all columns are proportional to a single vector), and $\|\cdot\|_F$ denotes the Frobenius norm.

Although aggregation propagates information across neighborhoods, successive linear and nonlinear transformations tend to compress feature diversity, resulting in rank collapse, often to a one-dimensional subspace Li et al. (2018); Oono & Suzuki (2019). In this process, all node embeddings become proportional to the leading eigenvector of $\tilde{A}$, erasing local distinctions and severely limiting

discriminative power Roth (2024); Liu et al. (2022); Keriven (2022). Moreover, conventional graph classification methods generate embeddings for every node and then apply global pooling. This pooling disregards hierarchical or substructure-specific organization, collapsing rich local cues into a single vector and further hampering classification accuracy Ying et al. (2018); Zhao et al. (2021); Alsentzer et al. (2020).

In this paper, we leverage perturbation theory and mutual information bounds to show that preserving embeddings of high-centrality node-induced subgraphs prevents rank collapse in deep MPNNs. The analysis characterizes how repeated neighborhood aggregation drives MPNNs toward a one-dimensional feature subspace, erasing local distinctions. Guided by these findings, we propose the subgraph plug-in (SP), a lightweight module that augments any base GNN, such as GCN or GAT, without modifying its architecture. SP computes centrality scores on the adjacency matrix to identify $\kappa$ key nodes (number of seeds) and extracts their $\tau$ hop neighborhoods (radius) as interpretable subgraphs, and concatenates the resulting embeddings with the global graph representation, thereby emphasizing subgraphs most predictive of the graph label while preserving original message-passing dynamics. Extensive experiments on 13 graph benchmarks, 13 GNN variants, 3 augmentation modules, 3 recent GNNs, and 5 subgraph-based methods demonstrate that SP consistently mitigates rank collapse and delivers superior classification and regression performance with a one-time preprocessing and negligible computational overhead.

## 2 RELATED WORK

### 2.1 AUGMENTATION METHODS FOR MITIGATING RANK COLLAPSE

Rank collapse (often to rank one) occurs when repeated message passing drives node features into a one-dimensional subspace Li et al. (2018); Oono & Suzuki (2019). Several strategies have been proposed to mitigate this collapse:

**Normalization and skip connections.** PairNorm adds a normalization step after each layer to preserve feature variance Zhao & Akoglu (2019). Jumping knowledge (JK) networks and GCNII introduce residual or identity mappings across layers to maintain embedding diversity Xu et al. (2018b); Chen et al. (2020). However, computational cost grows with depth, and fusing multi-layer features can dilute local signals.

**Stochastic graph structure removal.** DropEdge randomly drops edges during training to disrupt the fixed-point averaging that leads to collapse Rong et al. (2019). DropEdge injects variability but risks discarding task-critical links.

**Graph rewiring.** Graph rewiring methods adjust the adjacency matrix by adding or reweighting edges to shorten effective path lengths and improve signal propagation without altering the GNN's core update rule Attali et al. (2024); Banerjee et al. (2022). These methods require careful tuning and introduce additional computational overhead.

**Hierarchical and pooling schemes.** DiffPool learns soft cluster assignments to downsample the graph Ying et al. (2018), while TopKPool and SAGPool apply learnable pooling to retain a subset of nodes Lee et al. (2019). Although they can alleviate over-smoothing by hierarchical coarsening and node selection, they incur significant overhead and may erase fine-grained motifs.

These approaches remain within the standard message-passing paradigm, so mitigation can degrade as depth increases. In particular, several methods impose nontrivial hyperparameter tuning on a per-dataset basis, which complicates use in graph classification and generalization across diverse domains. Even when node features are well-separated, most graph classification pipelines compress them via simple readouts (sum, mean, max, or single-head attention) to a single graph vector. Such exchangeable pooling ignores the organization of multiple substructures and can further compress discriminative signals at the readout stage by averaging locally informative features. The methods above primarily target message passing and rarely redesign graph-level readout.

### 2.2 SUBGRAPH-BASED GNNS

Extracting subgraphs has become a popular strategy for boosting GNN expressivity and capturing higher-order structures: Union Subgraph GNNs generate node- or edge-deleted subgraphs to break

1-WL limitations Xu et al. (2024). MAG-GNN employs reinforcement learning to pick informative subgraphs, trading between expressivity and efficiency Kong et al. (2023). Policy-Learn uses two models, one to select a bag of subgraphs and another to predict Bevilacqua et al. (2023a). OSAN learns a distribution over tuples that represent subgraphs with multiple node markings Qian et al. (2022). CS-GNN associates subgraphs with node clusters rather than individual nodes to perform generalized message passing.

Although these models offer strong guarantees on expressivity, they often require intrusive architectural changes, dense tensor operations, or learned policy networks, which increase computational overhead and can introduce instability. Moreover, many subgraph GNNs construct a bag of candidate subgraphs per graph via random sampling, heuristics, or learned policies and then aggregate over this bag, which can miss domain-specific semantics and introduce additional pooling-induced information loss. By contrast, the proposed method computes simple node-centrality scores once, deterministically extracts only $\kappa$ subgraphs of radius $\tau$, and concatenates their embeddings with the global graph representation at readout, thereby preserving the base message-passing pipeline without architectural redesign, precomputed subgraph bags, or auxiliary selection networks.

## 3 PROPOSED METHOD

To mitigate rank collapse in deep message-passing neural networks, we introduce the subgraph plug-in (SP) in three stages:

1. **Structural analysis:** Compute centrality scores (degree, betweenness, closeness) on the input graph $\mathcal{G}$ to identify the top-$\kappa$ high-centrality nodes.

2. **Subgraph partitioning:** For each selected node $v$, extract its $\tau$-hop neighborhood as an interpretable subgraph $S_v$, yielding the collection $\mathbb{S} = \{S_v\}$; enforce disjointness among $\{S_v\}$ so that no nodes overlap, promoting independent information per subgraph and improving the utility of readout-time concatenation.

3. **Subgraph encoding and readout fusion:** Encode each subgraph $S \in \mathbb{S}$ with the same base GNN $f(\cdot; \boldsymbol{\theta})$ to obtain embeddings $\{\boldsymbol{h}_S\}$, then concatenate them with the global representation $\boldsymbol{h}_{\mathcal{G}}^{\text{base}}$ to form the final embedding $\boldsymbol{h}_{\mathcal{G}}$. This preserves the original message-passing flow while emphasizing locally discriminative structures at readout.

Figure 1 illustrates the process. We first detail the partitioning algorithm 1, then present theoretical guarantees, and finally describe embedding fusion and complexity.

### 3.1 PRELIMINARIES

Let $\mathcal{G} = (\mathbb{V}, \mathbb{E})$ be an undirected graph with $n = |\mathbb{V}|$ and adjacency matrix $\boldsymbol{A} \in \{0, 1\}^{n \times n}$. Let $d_{\mathcal{G}}(u, v)$ denote the shortest-path distance. We use three centrality measures:

**Definition 1 (Node centralities)** *For each node $v \in \mathbb{V}$,*

$$C_D(v) = \sum_{u \in \mathbb{V}} \boldsymbol{A}_{uv}, \quad C_B(v) = \sum_{\substack{s,t \in \mathbb{V} \\ s \neq t, \, v \notin \{s,t\}}} \frac{\sigma_{st}(v)}{\sigma_{st}}, \quad C_C(v) = \frac{n-1}{\sum_{u \in \mathbb{V}} d_{\mathcal{G}}(u,v)},$$

*where $\sigma_{st}$ is the number of shortest $s-t$ paths, $\sigma_{st}(v)$ counts those passing through $v$.*

(If $\mathcal{G}$ is disconnected, adopt harmonic closeness $C_C^{\text{harm}}(v) = \sum_{u \neq v} 1/d_{\mathcal{G}}(u,v)$ with $1/\infty = 0$.)

Let $f(\cdot; \boldsymbol{\theta})$ be the base GNN encoder and define the global representation by $\boldsymbol{h}_{\mathcal{G}}^{\text{base}} = \text{READOUT}(f(\mathcal{G}; \boldsymbol{\theta}))$.

### 3.2 SUBGRAPH PARTITIONING

We propose a partitioning scheme that concentrates on structurally informative regions. The method first computes multiple centrality measures (degree, betweenness, closeness) on the input graph $\mathcal{G}$, then expands $\tau$-hop neighborhoods around selected seeds. Each centrality emphasizes distinct aspects of structure (Appendix A.2). Neighborhood expansion ensures that resulting subgraphs

---

**Algorithm 1** Subgraph partitioning with disjoint $\tau$-hop neighborhoods

---

**Require:** graph $\mathcal{G} = (\mathbb{V}, \mathbb{E})$, centralities $\mathbb{C}$, seeds per measure $\kappa$, radius $\tau$
**Ensure:** subgraph collection $\mathbb{S}$

1: $\mathbb{S} \leftarrow \emptyset$, $\mathbb{U} \leftarrow \mathbb{V}$      // unassigned nodes
2: **for** each $c \in \mathbb{C}$ **do**
3:     compute scores $s_c(v)$ for all $v \in \mathbb{U}$
4:     $\mathbb{Q} \leftarrow$ nodes in $\mathbb{U}$ sorted by $s_c$
5:     $t \leftarrow 0$
6:     **for** each $v \in \mathbb{Q}$ **do**
7:       **if** $t = \kappa$ **then**
8:         **break**
9:       **end if**
10:      **if** $v \in \mathbb{U}$ **then**
11:        $B \leftarrow \{u \in \mathbb{U} : d_{\mathcal{G}}(u,v) \leq \tau\}$   // $\tau$-ball
12:        **if** $|B| > 0$ **then**
13:          $S_v \leftarrow B$; $\mathbb{S} \leftarrow \mathbb{S} \cup \{S_v\}$; $\mathbb{U} \leftarrow \mathbb{U} \setminus B$; $t \leftarrow t + 1$
14:        **end if**
15:      **end if**
16:     **end for**
17: **end for**
18: **return** $\mathbb{S}$

---

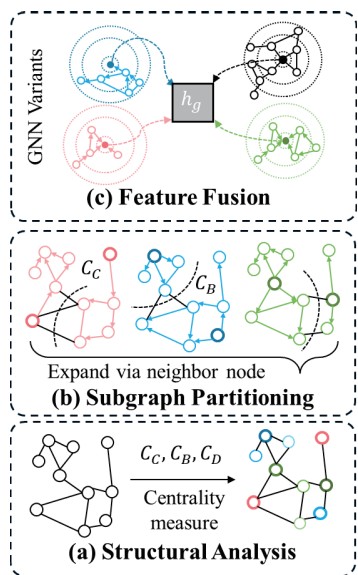

(c) Feature Fusion

(b) Subgraph Partitioning
Expand via neighbor node

(a) Structural Analysis

Figure 1: The overall process of SP

are readily encodable by GNNs with limited depth, reducing the risk that label-relevant nodes fall outside the receptive field. We enforce *disjoint* $\tau$-hop subgraphs by masking already-assigned nodes; when high-centrality seeds are adjacent, later seeds skip covered regions. The full procedure appears in Algorithm 1.

Focusing on these high-centrality regions aligns with the claim that localized structures around influential nodes preserve discriminative patterns. The information-theoretic formalization and its implications are developed in Section 3.3 below.

### 3.3 THEORETICAL GUARANTEES

Given a subgraph family $\mathcal{S}$, we study how focusing embeddings on $\mathcal{S}$ mitigates rank collapse—evidenced by non-decreasing pairwise dispersion under concatenation (Prop. 1) and by the information-monotonicity of the augmented representation (Lemma 1). The base encoder $f_\theta$ is fixed and deterministic. Subgraph sizes may vary; subgraphs are disjoint within a centrality type but may overlap across types. Further technical details are in Appendix A.

Let $\mathcal{S}^* = \bigcup_{S \in \mathcal{S}} S$. We use:

**A1 Task locality.** $Y \perp (G \setminus \mathcal{S}^*) \mid \mathcal{S}^*$.

**A2 Subgraph encoder adequacy.** There exists $\gamma \in (0,1]$ such that $I\big(H(G); Y\big) \geq \gamma\, I\big(\mathcal{S}^*; Y\big)$, where $H(G) = \text{Concat}\{h_S : S \in \mathcal{S}\}$.

**A3 Global contraction.** There exists $\eta \in [0,1]$ such that $I\big(f_{\text{global}}(G); Y\big) \leq \eta\, I\big(\mathcal{S}^*; Y\big)$.

**A2** reflects encoder sufficiency on $\mathcal{S}^*$ (e.g., bounded-depth subgraph encoders). **A3** abstracts spectral low-pass contraction in deep MPNNs; the contraction factor $\eta$ typically decays with depth as $|\lambda_2(\tilde{A})|^L$ Oono & Suzuki (2019). Empirically, our ablations, perturbation tests, and depth-sensitivity studies collectively corroborate **A1–A3**.

**Theorem 1 (Information comparison under locality and contraction)** *Under A1–A3, if $\gamma \geq \eta$ then*

$$I\big(H(G); Y\big) \;\geq\; I\big(f_{\text{global}}(G); Y\big), \tag{3}$$

*with strict inequality when $\gamma > \eta$.*

By **A2**, $I(H(G); Y) \geq \gamma I(\mathcal{S}^*; Y)$. By **A3**, $I(f_{\text{global}}(G); Y) \leq \eta I(\mathcal{S}^*; Y)$. If $\gamma \geq \eta$, the claim follows.

**Corollary 1 (Depth advantage from spectral decay; qualitative)** *If the base global pipeline exhibits oversmoothing so that for some $c \in (0, 1)$ and large $L$, $I(f_{\text{global}}(G); Y) \leq c^L I(\mathcal{S}^*; Y)$ (e.g., governed by $|\lambda_2(\tilde{A})|^L$), while $H(G)$ is computed at bounded depth, then there exists $L_0$ such that for all $L \geq L_0$, $I(H(G); Y) > I(f_{\text{global}}(G); Y)$.*

Deep MPNNs contract node distinctions geometrically with depth due to low-pass behavior and spectral gap; see Li et al. (2018); Oono & Suzuki (2019); Chen et al. (2020).

**Proposition 1 (Graph-level dispersion is nondecreasing under concatenation)** *For any two graphs $G_1, G_2$, let $d_{\text{base}} = \| h_{G_1}^{\text{base}} - h_{G_2}^{\text{base}} \|_2$ and $d_{\text{SP}} = \big\| [h_{G_1}^{\text{base}} \oplus H(G_1)] - [h_{G_2}^{\text{base}} \oplus H(G_2)] \big\|_2$. Then $d_{\text{SP}} \geq d_{\text{base}}$. Consequently, any dataset-level pairwise (Euclidean) dispersion of graph embeddings is nondecreasing after adding $H(G)$.*

By Pythagorean expansion in the augmented coordinates.

**Lemma 1 (Augmented representation is information-monotone)** *Let $f_{\text{global}}(G)$ and $H(G)$ be deterministic. Then*

$$I\big([f_{\text{global}}(G), H(G)]; Y\big) = I\big(f_{\text{global}}(G); Y\big) + I\big(H(G); Y \mid f_{\text{global}}(G)\big) \geq I\big(f_{\text{global}}(G); Y\big).$$

Follows from the chain rule and nonnegativity of conditional mutual information.

**Proposition 2 (Perturbation sensitivity)** *Let $G' = P_\alpha(G, \mathcal{S}^*)$ and $G'' = P_\alpha(G, G \setminus \mathcal{S}^*)$ denote $\alpha$ i.i.d. random edits restricted inside vs. outside $\mathcal{S}^*$. Let $\phi$ be the trained predictor. Suppose each edit inside (resp. outside) flips the predicted label with probability $p$ (resp. $q$), with $p > q$. Then*

$$\Pr[\phi(G') \neq \phi(G)] - \Pr[\phi(G'') \neq \phi(G)] = (1-q)^\alpha - (1-p)^\alpha \geq \alpha(p-q)(1-p)^{\alpha-1} > 0.$$

The equality uses independence; the inequality follows from the mean-value theorem applied to $g(r) = (1-r)^\alpha$ with $r \in (q, p)$ (For small $p, q$, $(1-r)^\alpha \approx e^{-\alpha r}$ gives the familiar approximation $e^{-\alpha q} - e^{-\alpha p}$).

## 3.4 FEATURE FUSION

Having established the rationale, we realize SP via readout-time fusion. Each (within-type disjoint) subgraph is encoded to preserve local structure and then integrated with the global embedding, ensuring that partition-specific signals are not washed out by a single global aggregator.

Let $f(\cdot; \theta)$ be the base GNN encoder and READOUT a graph-level pooling. Compute

$$h_G^{\text{base}} = \text{READOUT}\big(f(G; \theta)\big), \qquad h_S = \text{READOUT}\big(f(G[S]; \theta)\big) \text{ for } S \in \mathcal{S}, \qquad (4)$$

and fuse

$$h_G = \text{MLP}\Big([h_G^{\text{base}} \oplus \{h_S\}_{S \in \mathcal{S}}]\Big). \qquad (5)$$

This preserves the base message-passing pipeline while injecting localized, high-centrality signals. The extra cost scales with $|\mathcal{S}|$ feed-forwards through $f$, whereas centrality scoring/partitioning is a one-time preprocessing step.

## 3.5 COMPLEXITY ANALYSIS

Let $n$ be the number of nodes, $m$ edges, $d$ embedding width, $\tau$ hop radius, and $\kappa = |\mathcal{S}|$ selected subgraphs. The main costs are:

**Centrality scoring:** degree $\mathcal{O}(n+m)$, betweenness (Brandes, unweighted) $\mathcal{O}(nm)$

**$\tau$-hop extraction:** $\mathcal{O}\Big(\sum_{S \in \mathcal{S}}(n_S + m_S)\Big) \leq \mathcal{O}\big(\min\{\kappa(n+m), \kappa \bar{d}^\tau\}\big)$,

**Fusion (concatenate+MLP):** $\mathcal{O}(d^2 + \kappa d)$.

Table 1: Performance comparison on four chemical/biological datasets *with* node features. "Original" indicates baseline GNN accuracy, and "Ours" refers to the proposed method accuracy. Results are mean±std (%).

| METHOD | MUTAG | | NCI1 | | NCI109 | | PROTEINS | |
|---|---|---|---|---|---|---|---|---|
| | ORIG. | OURS | ORIG. | OURS | ORIG. | OURS | ORIG. | OURS |
| GCN | 74.0±6.1 | **76.6±5.1** | 69.2±2.7 | **71.8±5.2** | 67.5±2.9 | **68.9±3.4** | 71.6±3.8 | **74.1±4.3** |
| GIN | 81.0±10.2 | **82.5±9.9** | 76.4±4.1 | **77.2±4.9** | 73.6±5.2 | **74.7±4.1** | 71.4±4.4 | **72.2±4.0** |
| GIN0 | 80.9±7.5 | **82.3±12.2** | 74.9±3.7 | **76.8±3.0** | 75.0±2.9 | **75.1±2.7** | 70.1±4.1 | **71.9±3.9** |
| TOPK | 72.9±5.8 | **78.2±5.6** | 71.6±4.6 | **74.9±3.9** | 69.9±3.2 | **71.7±2.9** | 72.0±3.4 | **72.1±4.2** |
| SAGPOOL | 80.8±10.8 | **83.0±7.4** | 70.8±5.1 | **74.5±7.9** | 70.9±3.6 | **72.0±3.2** | 71.8±3.4 | **72.2±5.4** |
| EDGEPOOL | 73.5±5.9 | **78.9±9.5** | **73.1±2.5** | 72.4±5.3 | 70.1±5.6 | **70.4±2.6** | 71.0±3.6 | **71.3±1.4** |
| GRACLUS | 77.1±5.9 | **77.3±7.3** | 71.3±4.4 | **76.0±1.7** | 70.3±2.9 | **71.2±3.6** | 71.9±3.3 | **72.5±3.2** |
| GAT | 75.5±8.9 | **77.1±8.0** | 71.5±4.8 | **72.2±5.6** | 68.5±4.7 | **68.7±4.4** | 71.8±4.0 | **72.8±4.3** |
| SET2SET | **73.4±11.4** | 70.3±12.1 | 70.4±3.5 | **73.8±3.9** | 79.7±2.7 | **80.2±1.3** | 73.1±4.7 | **75.1±2.7** |
| GRAPHSAGE | 77.2±4.7 | **79.5±3.2** | 71.3±3.5 | **76.1±3.1** | 69.6±3.1 | **70.3±3.2** | 71.6±2.7 | **71.8±3.2** |

**Dominant step.** In practice, the most time-consuming component is *centrality scoring* when using betweenness/closeness, i.e., $\mathcal{O}(nm)$ (unweighted) or $\mathcal{O}(nm+n^2 \log n)$ (weighted). Subgraph extraction is linear in the total size of selected subgraphs, and fusion is negligible.

This preprocessing is one-time per graph (cacheable) and $\kappa \ll n$ in our settings. The per-epoch training overhead from processing $\kappa$ subgraphs is roughly depth-invariant (both base and SP passes scale similarly with added layers). Empirical timings are reported in Appendix C.4 and C.5.

## 4 EXPERIMENTAL RESULTS

### 4.1 DATASETS AND EXPERIMENT SETUP

**Dataset.** We evaluate the proposed method on 13 benchmark datasets commonly used in graph classification. These datasets span social networks (IMDB-BINARY, IMDB-MULTI, COLLAB), chemical compounds (MUTAG, PTC, NCI1, NCI109), proteins (PROTEINS), and large-recent chemical dataset (ogbg-molhiv, ZINC) More description is in Appedix B.1.

**Training setup.** To verify the best performance of all GNNs used in the experiments for each dataset, we have performed experiments with five different numbers of layers and dimensions, running each setting for 200 epochs in 10-fold cross-validation. For fair comparison, all experiments on Table 1–3 are conducted by setting the $\kappa$ for extracting the subgraph to 2, and $\tau$ for expanding the subgraph to 4. The resulting experimental outcomes and hyperparameters are detailed in Appendix B.2. We compare the proposed method against 13 baseline approaches, 3 augmentation modules, 5 subgraph-based methods, and 3 recent GNNs, covering standard GNN models, pooling strategies, and enhancement modules (see Appendix B.3 for full details).

### 4.2 ACCURACY COMPARISON OF SP ON GNN VARIANT

Tables 1–3 compare the classification accuracies of various GNN baselines (*"Original"*) and their SP-augmented counterparts (*"Ours"*). The key finding is that SP consistently improves graph classification accuracy in 96 of the 110 experimental configurations. In particular, even for models such as GIN, which already exhibit competitive performance, SP reliably increases accuracy across most datasets.

Interestingly, the most pronounced gains occur on large graphs. For example, on COLLAB and IMDB-BINARY (Table 3), SP yields absolute accuracy gains of 2–3 pp. This pattern is consistent with our analysis that deep architectures on large graphs are prone to rank-one collapse; by emphasizing critical local information that would otherwise be washed out by smoothing, SP mitigates this effect. Consequently, SP functions as a lightweight, architecture-agnostic plug-in that preserves high-centrality substructures without modifying the underlying GNN.

Table 2: Performance comparison on four PTC-family datasets *with* node features. Same legend as Table 1.

| METHOD | PTC_MR | | PTC_FR | | PTC_MM | | PTC_FM | |
|---|---|---|---|---|---|---|---|---|
| | ORIG. | OURS | ORIG. | OURS | ORIG. | OURS | ORIG. | OURS |
| GCN | 54.9±5.0 | **56.4±6.3** | 65.5±2.9 | **66.9±2.5** | 66.6±8.4 | **67.5±5.6** | 60.7±3.9 | **61.5±4.7** |
| GIN | 55.6±4.1 | **56.1±3.1** | 64.1±5.7 | **64.5±3.7** | 62.2±7.7 | **65.1±7.1** | 61.0±4.4 | **61.7±5.2** |
| GIN0 | 56.4±6.6 | **58.1±6.2** | 64.4±7.3 | **65.8±4.1** | 64.0±3.2 | 59.5±6.2 | **60.2±2.9** | 58.2±7.5 |
| TOPK | 57.3±8.8 | **57.5±8.3** | 66.1±4.2 | **66.7±4.7** | 66.9±4.8 | 66.6±6.3 | 60.2±3.9 | **61.3±7.6** |
| SAGPOOL | 57.2±6.2 | **57.9±5.0** | 66.1±4.8 | **67.2±3.7** | 66.0±5.2 | **67.5±5.3** | 60.7±4.5 | **61.2±6.6** |
| EDGEPOOL | 55.2±7.4 | **56.1±6.9** | 64.7±7.1 | **65.2±4.6** | 65.4±6.2 | **67.8±5.1** | 60.2±4.8 | **60.7±5.6** |
| GRACLUS | 52.9±9.4 | **56.7±4.6** | 65.5±5.7 | 65.2±4.6 | 64.8±6.1 | **66.0±5.4** | 61.0±4.4 | **63.3±3.9** |
| GAT | 53.2±7.0 | **58.4±5.6** | 65.2±5.9 | **65.3±5.2** | 67.8±6.0 | **69.0±5.7** | 59.6±4.0 | **61.3±4.2** |
| SET2SET | **55.2±4.5** | 53.8±6.8 | 66.7±3.9 | **66.9±3.2** | 67.8±5.7 | **68.3±1.4** | 59.9±4.3 | **61.4±1.9** |
| GRAPHSAGE | 54.1±6.1 | **56.7±9.0** | 66.4±2.2 | 63.5±6.0 | 63.4±5.7 | **66.9±5.8** | 60.5±4.1 | 59.9±8.2 |

Table 3: Performance comparison on the datasets *without* node features (IMDB-BINARY, IMDB-MULTI, COLLAB). Again, "Original" vs. "Ours" refers to baseline vs. proposed method. Results are mean±std (%).

| METHOD | IMDB-BINARY | | IMDB-MULTI | | COLLAB | |
|---|---|---|---|---|---|---|
| | ORIG. | OURS | ORIG. | OURS | ORIG. | OURS |
| GCN | 74.6±4.8 | **74.9±3.8** | 50.9±3.6 | 50.9±3.5 | 80.6±0.4 | **82.1±0.9** |
| GIN | 72.7±5.5 | **73.4±5.7** | 48.7±3.0 | **50.6±2.9** | 80.2±0.6 | **81.2±0.8** |
| GIN0 | 73.6±4.4 | **73.7±5.4** | 48.3±2.3 | **49.9±4.3** | 80.1±0.2 | **81.2±1.7** |
| TOPK | 74.3±5.6 | **75.9±5.3** | 50.5±2.7 | **50.7±3.0** | 78.8±1.4 | **79.1±2.6** |
| SAGPOOL | 73.4±5.3 | **74.1±5.4** | 50.5±2.5 | **50.6±3.4** | 80.3±1.2 | **81.6±1.0** |
| EDGEPOOL | **73.5±5.0** | 73.2±4.7 | 50.8±2.5 | **51.5±4.9** | 82.4±1.0 | **83.8±0.7** |
| GRACLUS | 72.7±4.7 | **73.2±3.1** | 50.7±2.9 | **50.8±2.7** | 79.4±0.5 | 78.8±1.1 |
| GAT | **73.0±4.6** | 72.4±4.2 | 50.0±3.2 | **50.9±2.0** | 80.3±0.7 | **81.8±2.1** |
| SET2SET | **72.9±4.4** | 72.5±4.9 | 50.9±2.9 | **51.2±1.7** | 77.6±0.8 | 76.4±3.1 |
| GRAPHSAGE | 71.7±3.9 | **72.8±3.6** | 51.2±3.0 | **52.3±1.7** | **79.6±2.0** | 78.6±1.7 |

Although SP appears to underperform in 16 configurations across Tables 1–3 or to produce gains within one standard deviation of the baseline, this follows from strict adherence to each model's original hyperparameters without any SP-specific tuning. A detailed error analysis is provided in Section C.1, and the hyperparameters used in all experiments are listed in Table 8. SP is designed to counter rank-one collapse that intensifies with depth; in our experiments, deeper architectures augmented with SP outperform the untuned baselines. These gains arise without additional hyperparameter optimization, indicating robustness and practical utility. Section 4.3 and Appendix C show that stacking more layers further amplifies SP's benefits.

### 4.3 ACCURACY COMPARISON WITH RECENT METHODS

Across OGBG–MolHIV and ZINC, the proposed method is the only augmentation that yields consistent gains across backbones and delivers the best ZINC MAE for every encoder; on MolHIV it is strongest for three of four backbones, with GraphSAGE + PairNorm as the lone exception. Head–to–head against recent SubGNNs at $T=3(\kappa = 1)$ (GIN backbone), it attains the top score on both benchmarks, indicating that a small, deterministically chosen set of $\kappa$ radius $\tau$ subgraphs can outperform bag–based subgraph pipelines. These outcomes support the theory that fusing locally focused subgraph embeddings at readout preserves label–relevant structure while avoiding architectural changes to the base MPNN. Detailed error analysis appears in Sec. C.1; hyperparameters are listed in Tab. 8.

Notably, GCNII and U-Net exhibit poor performance when evaluated in these domains. This degradation is not merely due to limited representational capacity of the GNN encoder but stems from

Table 4: Comparison of additional methods on various GNN models (OGBG-MolHIV, ZINC). T=3 results for SGNN baselines are appended.

| MODEL | ADDITIONAL METHOD | OGBG(ROC-AUC) | ZINC(MAE) |
|---|---|---|---|
| GCN | BASE | $0.62788 \pm 0.00451$ | $0.14852 \pm 0.01511$ |
| | DROP | $0.62318 \pm 0.00387$ | $0.14655 \pm 0.01248$ |
| | PAIRNORM | $0.53268 \pm 0.00671$ | $0.14877 \pm 0.01346$ |
| | WITHJK | $0.67570 \pm 0.00299$ | $0.14256 \pm 0.00988$ |
| | **SP** | $\mathbf{0.69194 \pm 0.00257}$ | $\mathbf{0.13159 \pm 0.00913}$ |
| GIN | BASE | $0.80840 \pm 0.01581$ | $0.07125 \pm 0.00594$ |
| | DROP | $0.76590 \pm 0.01266$ | $0.08349 \pm 0.00581$ |
| | PAIRNORM | $0.77442 \pm 0.01007$ | $0.08158 \pm 0.00563$ |
| | WITHJK | $0.81132 \pm 0.00864$ | $0.07345 \pm 0.00601$ |
| | **SP** | $\mathbf{0.84810 \pm 0.00912}$ | $\mathbf{0.07011 \pm 0.00544}$ |
| GIN0 | BASE | $0.81036 \pm 0.00457$ | $0.08641 \pm 0.00648$ |
| | DROP | $0.76776 \pm 0.00866$ | $0.08358 \pm 0.00650$ |
| | PAIRNORM | $0.77776 \pm 0.00948$ | $0.08415 \pm 0.00684$ |
| | WITHJK | $0.81136 \pm 0.00900$ | $0.08478 \pm 0.00631$ |
| | **SP** | $\mathbf{0.83892 \pm 0.01012}$ | $\mathbf{0.07861 \pm 0.00694}$ |
| GRAPHSAGE | BASE | $0.72514 \pm 0.01022$ | $0.06849 \pm 0.00812$ |
| | DROP | $0.66738 \pm 0.01051$ | $0.06579 \pm 0.00791$ |
| | PAIRNORM | $0.79502 \pm 0.01069$ | $0.06661 \pm 0.00755$ |
| | WITHJK | $0.68638 \pm 0.00998$ | $0.06922 \pm 0.00764$ |
| | **SP** | $\mathbf{0.75094 \pm 0.01002}$ | $\mathbf{0.06294 \pm 0.00734}$ |
| GCNII* | NONE | $0.49914 \pm 0.05121$ | $2.64234 \pm 0.56150$ |
| UNET* | NONE | $0.45804 \pm 0.04966$ | $1.44671 \pm 0.49660$ |
| GRAPHORMER | NONE | $0.79392 \pm 0.00358$ | $0.05648 \pm 0.00258$ |
| | **SP** | $\mathbf{0.83311 \pm 0.00426}$ | $\mathbf{0.04951 \pm 0.00322}$ |
| GIN (SUBGNN, T=3) | RANDOM | $0.76300 \pm 0.01000$ | $0.11200 \pm 0.00600$ |
| | MAG-GNN | $0.80600 \pm 0.01900$ | $0.11000 \pm 0.01200$ |
| | OSAN | $0.77400 \pm 0.02100$ | $0.19400 \pm 0.00600$ |
| | POLICY-LEARN | $0.83500 \pm 0.01500$ | $0.09100 \pm 0.00600$ |
| | CS-GNN | $0.79600 \pm 0.01900$ | $0.09300 \pm 0.00700$ |
| | HYMN | $0.83700 \pm 0.02100$ | $0.09000 \pm 0.00600$ |
| | **SP (OURS)** | $\mathbf{0.83800 \pm 0.02100}$ | $\mathbf{0.08600 \pm 0.00500}$ |

information loss during the graph pooling step, where independent node embeddings are aggregated into a single vector. SP addresses this challenge by simultaneously preventing rank-one collapse and employing feature fusion to recover and integrate critical local information that would otherwise be discarded.

## 4.4 ABLATION & PERTURBATION ANALYSIS

**Setup.** Let $\mathbb{S}^* = \bigcup_{S \in \mathbb{S}} S$. We test (A1) *Task locality*—$Y \perp (\mathcal{G} \setminus \mathbb{S}^*) \mid \mathbb{S}^*$—via node–feature masking, and (A2) *Subgraph encoder adequacy*—$I(H(\mathcal{G}); Y) \geq \gamma I(\mathbb{S}^*; Y)$—via centrality–choice ablations.

**A1: Node–feature masking (perturbation).** For each dataset–backbone pair, we compare **Random** masking (uniform nodes) vs. **Centrality** masking (top nodes by degree/betweenness/closeness) at the same masking rate; results are in Table 5. Across all 30/30 valid comparisons, accuracy under **Centrality** masking is lower than under **Random** masking (two runs failed to train when central nodes were masked, marked "–"), indicating that information critical for $Y$ is concentrated in $\mathbb{S}^*$ rather than $\mathcal{G} \setminus \mathbb{S}^*$. This selective fragility directly supports (A1).

**A2: Centrality–choice ablation (encoder adequacy).** Using a GCN backbone, we replace SP's selection with a *single* centrality (degree, betweenness, closeness) and report performance in Table 6. Any single centrality strictly improves over the base in all datasets, and **SP (ours)**—which fuses

Table 5: Comparison of graph classification performance under node feature masking. '-' means that the model failed to train properly

| | GCN | | GAT | | GIN | | SET2SET | |
|---|---|---|---|---|---|---|---|---|
| DATASET | RND. | CENT. | RND. | CENT. | RND. | CENT. | RND. | CENT. |
| MUTAG | **72.7** | 68.7 | **71.9** | 66.3 | **78.4** | 76.6 | **70.4** | 69.4 |
| NCI1 | **64.9** | 63.5 | **67.6** | 65.5 | **73.7** | 71.1 | **68.7** | 65.8 |
| NCI109 | **63.8** | 62.2 | **65.4** | 64.2 | **70.6** | 67.8 | **75.6** | 71.4 |
| PROTEINS | **65.7** | 60.7 | **67.9** | 63.7 | **67.8** | 64.9 | **71.9** | 69.8 |
| PTC_MR | **52.6** | – | **50.1** | – | **50.9** | 50.3 | **52.4** | 51.2 |
| PTC_FR | **63.0** | 62.7 | **59.7** | 58.6 | **60.3** | 60.2 | **63.6** | 61.2 |
| PTC_MM | **62.8** | 60.3 | **62.2** | 61.0 | **57.1** | 54.9 | **61.8** | 60.2 |
| PTC_FM | **57.9** | 55.8 | **55.5** | 53.9 | **53.7** | 52.7 | **57.3** | 56.6 |

Table 6: **Ablation on centrality measures** (GCN backbone). Higher is better for COLLAB, MUTAG, PROTEINS, OGBG−MolHIV; lower is better for ZINC. Mean ± std over repeated runs.

| Dataset | GCN base | Degree | Betweenness | Closeness | SP (ours) |
|---|---|---|---|---|---|
| COLLAB | 80.6 ± 0.4 | 81.9 ± 0.5 | 81.3 ± 0.6 | 81.7 ± 0.7 | 82.1 ± 0.9 |
| MUTAG | 74.0 ± 6.1 | 74.8 ± 5.5 | 75.5 ± 5.3 | 74.1 ± 5.2 | 76.6 ± 5.1 |
| PROTEINS | 71.6 ± 3.8 | 72.2 ± 3.9 | 73.0 ± 4.0 | 73.6 ± 4.2 | 74.1 ± 4.3 |
| OGBG−MolHIV (AUC) | 0.627 ± 0.005 | 0.643 ± 0.006 | 0.685 ± 0.007 | 0.686 ± 0.008 | 0.692 ± 0.003 |
| ZINC (MAE) | 0.149 ± 0.015 | 0.145 ± 0.014 | 0.137 ± 0.013 | 0.135 ± 0.013 | 0.132 ± 0.009 |

$\kappa$ radius–$\tau$ subgraph embeddings—achieves the best score in every case (e.g., OGBG−MolHIV $0.6279 \rightarrow 0.6919$ ROC–AUC; ZINC $0.1485 \rightarrow 0.1316$ MAE). These gains show that $H(\mathcal{G}) = \text{Concat}\{h_S : S \in \mathbb{S}\}$ retains label–relevant signals present in $\mathbb{S}^*$, substantiating (A2) and aligning with Theorem 1 on the augmented representation's information advantage.

## 4.5 LIMITATIONS AND SCOPE

**(i) Tie handling and invariance.** Deterministic tie-breaking by node index can, in principle, violate permutation invariance. In practice we observed negligible effect; nevertheless, a canonical, isomorphism-invariant rule (e.g., WL colors with lexicographic neighborhood hashes) or a fixed hash of local neighborhoods can resolve ties without sacrificing invariance.

**(ii) Tasks driven by global structure.** When labels depend on long-range global topology, the marginal gain from local subgraphs may shrink. SP is orthogonal to rewiring and positional encodings and can be paired with them to restore long-range signal.

**(iii) Very large graphs.** Exact betweenness/closeness can be costly. Approximate centralities (e.g. PageRank proxies) are drop-in replacements; centrality is computed once and reused across $(\kappa, \tau)$ sweeps.

## 5 CONCLUSION

We propose *Subgraph Plug-in* (SP), a lightweight, architecture-agnostic module that (1) scores nodes via simple centralities, (2) extracts $\kappa$ radius-$\tau$ subgraphs, and (3) concatenates their embeddings with the global graph representation. The perturbation and information-theoretic analyses explain why emphasizing high-centrality neighborhoods counters depth-induced contraction (rank-one collapse), and the information-monotonicity result guarantees that fusion does not decrease label information. Empirically, SP yields consistent gains across 13 datasets and 11 backbones, with the largest improvements on large graphs and deeper models, and remains competitive with recent SGNNs at modest cost. Future works include adaptive selection of $(\kappa, \tau)$, scalable approximate centralities for web-scale graphs and joint use with rewiring or positional encodings. We view SP as a practical, last-layer structural prior: a simple partition–encode–fuse step that reliably preserves discriminative substructures while leaving the base MPNN pipeline intact.

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
