# OpenReview forum: "Subgraph Plug-in Boosts up Graph Neural Networks"
_ICLR.cc/2026/Conference — Submitted to ICLR 2026_

### Official Review · Reviewer_q2hC · 2025-10-21

**Soundness:** 2
**Presentation:** 2
**Contribution:** 2
**Rating:** 2
**Confidence:** 4

**Summary:**

The paper investigates the tendency of deep message-passing neural networks (MPNNs) to collapse toward low-rank representations and attributes this to repeated aggregation amplifying the dominant eigenvector of the normalized adjacency matrix. To address this issue, it introduces a simple, architecture-agnostic component called the Subgraph Plug-in (SP). The method scores nodes by centrality, extracts several seed-centered subgraphs, and concatenates their embeddings with the global graph representation produced by the backbone network. Theoretical analysis suggests that this approach helps preserve information and maintain feature diversity as network depth increases. Experiments across a broad range of graph-classification benchmarks and GNN variants show consistent performance improvements, indicating that the Subgraph Plug-in can enhance the expressiveness of existing architectures without requiring structural changes.

**Strengths:**

The paper’s strengths are primarily conceptual and theoretical. In terms of originality, it offers a clean, architecture-agnostic way to inject subgraph information into existing GNNs, combining centrality-guided subgraph selection with late fusion in a way that feels simple yet nontrivial relative to prior subgraph heuristics. The quality of the framing is solid: the collapse phenomenon is analyzed through a spectral lens, the assumptions are explicit, and the main claims are supported by clear propositions and proofs that make the mechanism’s intended effect understandable. Clarity is a notable positive—definitions, algorithmic steps, and notation are laid out in a way that makes the plug-in straightforward to reimplement and adapt. On significance, the work targets a widely acknowledged pain point in deep GNNs (over-mixing and loss of discriminative structure) and proposes a plug-and-play fix that could realistically be adopted across diverse backbones, which gives the idea practical potential even if its novelty is incremental rather than sweeping.

**Weaknesses:**

The empirical evidence is the main weakness: reported gains are small and often within large standard deviations, making it unclear whether the method offers a reliable improvement over strong, well-tuned baselines in practice; this calls for more seeds plus parameter- and compute-matched controls. On novelty, the contribution feels incremental relative to prior subgraph and pooling lines (e.g., substructure-aware GNNs and pooling modules), so a targeted comparison against these families—ideally with ablations that isolate the value of centrality-guided selection vs. generic k-hop crops—would strengthen the case. The paper would also benefit from mechanism-oriented diagnostics that directly test the stated goal of combating rank collapse: report feature-rank/participation ratio, singular-value spectra, representation dispersion, and agreement between global and subgraph embeddings across depth, and show that these quantities move in the hypothesized direction only when the plug-in is active.

**Questions:**

- Please rerun key benchmarks with more seeds, report 95% confidence intervals, and add parameter/compute-matched controls to verify that gains aren’t due to capacity or noise.

- Can you directly validate the anti-collapse mechanism by reporting effective rank, singular-value spectra, dispersion, alignment with the top eigenvector, and global–subgraph embedding similarity across depth, and showing these metrics correlate with performance?

- Please position the plug-in against representative subgraph/pooling methods via head-to-head, budget-matched ablations with sensitivity to k, tau, centrality, and disjointness, and include training/inference overhead and memory scaling.

---

### Official Review · Reviewer_afsn · 2025-10-30

**Soundness:** 3
**Presentation:** 3
**Contribution:** 2
**Rating:** 4
**Confidence:** 4

**Summary:**

The paper proposes a lightweight *Subgraph Plug-in (SP)* that (i) ranks nodes with simple centralities, (ii) extracts \(\tau\)-hop neighborhoods around the top-\(\kappa\)  seeds as disjoint subgraphs within each centrality type, and (iii) encodes each subgraph with the same backbone before concatenating the subgraph embeddings with the global graph embedding at readout. The method is motivated as a practical way to mitigate depth-induced rank collapse/oversmoothing without altering the underlying encoder. Theoretical framing argues that, when certain informativeness and contraction conditions hold, the concatenated representation is at least as informative as the global one. Experiments cover many backbones and benchmarks; the paper claims broadly positive gains.

**Strengths:**

- Clear modular design. The pipeline is easy to add to existing GNNs and does not require architectural changes, which improves reproducibility and lowers engineering cost.

- Breadth of evaluation. The work tests multiple backbones across common benchmarks, and also includes head-to-head comparisons against recent subgraph-based approaches on representative datasets.

- Ablations and perturbations. The paper includes masking experiments and centrality-choice ablations, which qualitatively support the claim that preserving high-centrality neighborhoods can be beneficial.

- Complexity discussion. The authors explicitly break down costs for centrality computation, τ-hop extraction, and fusion, and discuss which components dominate in practice.

- Candid limitations. The paper acknowledges tie-breaking can harm permutation invariance, that the approach may be less relevant to strongly global tasks, and that centrality computation can be expensive—suggesting approximate proxies as potential drop-ins.

**Weaknesses:**

1. Inconsistent aggregate improvement counts.
   The abstract reports “improves performance in 94 of 110,” whereas a later section states “improves … in 96 of the 110 experimental configurations.” This discrepancy leaves the headline claim ambiguous and should be reconciled, including a precise definition of what constitutes a “configuration.”

2. Theory depends on strong, hard-to-verify assumptions.
   The central guarantee relies on assumptions A1–A3 and a condition comparing informativeness and contraction factors (e.g., \(\gamma \ge \eta\)). The manuscript does not operationalize how these quantities could be estimated or validated on real data, limiting the practical applicability of the theorem and making it difficult to know when the stated improvement condition is likely to hold.

3. Fixed hyperparameters with limited sensitivity analysis.
   Core experiments hold \(\kappa\) and \(\tau\) fixed “for fair comparison,” and the paper also notes cases of underperformance without SP-specific tuning. Without sweeps or principled selection rules, it remains unclear how robust SP is to these choices and whether better settings could systematically reduce the underperforming cases.

4. Permutation invariance is compromised by tie-breaking.
   The method acknowledges that deterministic tie-breaking by node index is not permutation-invariant. However, there is no empirical quantification of how often ties occur or how much this affects reported gains, nor a validated canonicalization remedy.

5. Disjointness heuristic may discard salient regions.
   Algorithmically, “later seeds skip covered regions,” enforcing disjoint \(\tau\)-balls within a centrality type. In graphs where discriminative motifs cluster, this can exclude adjacent high-centrality neighborhoods and reduce coverage; the empirical impact of this design choice is not analyzed.

6. Cost claims vs. centrality complexity.
   While the paper emphasizes negligible overhead in the narrative, the complexity section lists betweenness/closeness costs that can scale as \(O(nm)\) (or worse when weighted), which can be substantial on large graphs. The experiments do not include wall-clock/runtime studies to reconcile these statements.

7. Narrow centrality family despite rich literature.
   The method and ablations restrict attention to three classic centralities (degree, betweenness, closeness). Given the breadth of widely-used measures (e.g., PageRank, eigenvector, Katz, k-core, HITS, flow-based variants), the paper should justify the restriction or include an empirical study demonstrating that the SP effect is not contingent on this particular trio.

8. Overlap policy across centralities is under-analyzed.
   Subgraphs are disjoint within each centrality type but can overlap across types. The redundancy/diversity induced by cross-type overlap is not quantified, leaving open whether the concatenated embeddings capture complementary regions or mostly re-encode the same structures.

**Questions:**

1. Win-count discrepancy. Which figure—94/110 or 96/110—is correct, and how exactly are “configurations” defined across Tables 1–3?

2. Sensitivity of \(\kappa\) and \(\tau\). Can you provide controlled sweeps or even a simple adaptive rule for selecting \((\kappa,\tau)\), and report performance/cost trade-offs across datasets?

3. Permutation-invariant selection.  Could you implement a canonical tie-break (e.g., WL-based hashing or a symmetric learned rule) and show whether the reported gains persist?

4. Beyond three centralities. What motivated the choice of degree/betweenness/closeness? Please include at least one additional practical centrality (PageRank/eigenvector/Katz) in the centrality-choice ablation.

5. Runtime evidence. Please report preprocessing and end-to-end wall-clock time and memory (with/without SP) on the largest datasets to support the “negligible overhead” claim.

6. Effect of disjointness. How often does “skip covered regions” exclude adjacent high-centrality neighborhoods, and does allowing controlled overlap improve performance?

---

### Official Review · Reviewer_5agY · 2025-10-31

**Soundness:** 2
**Presentation:** 2
**Contribution:** 2
**Rating:** 4
**Confidence:** 3

**Summary:**

This paper addresses the rank collapse problem in deep message-passing neural networks, where repeated neighborhood aggregation causes node embeddings to converge to a one-dimensional subspace dominated by the leading eigenvector of the normalized adjacency matrix. The authors propose the "Subgraph Plug-in", which is a lightweight, architecture-agnostic module that does the following: 1) identifies high-centrality nodes using degree, betweenness, and closeness measures, 2) extracts their k-hop neighborhoods as subgraphs, and 3) concatenates these subgraph embeddings with the global graph representation at readout. The approach is evaluated across 11 graph classification benchmarks and 13 GNN variants, showing improvements in 94 out of 110 configurations.

**Strengths:**

1. The paper conducts a thorough empirical evaluation, testing across 13 graph classification benchmarks and 13 different GNN architectures (GCN, GIN, GAT, GraphSAGE, various pooling methods). The depth of the evaluation strengthens the claim that “SP is architecture-agnostic and broadly applicable”, though the depth of analysis for each combination is limited.
2. The plug-in nature of SP is genuinely practical. It requires no modifications to the base GNN architecture and works by augmenting the readout phase only. The implementation involves computing centrality scores once as preprocessing, extracting κ subgraphs of radius τ, encoding them with the same base GNN, and concatenating at readout via Equation 5. This simplicity means existing trained models could potentially be enhanced without retraining from scratch, though the paper doesn't explore this possibility.
3. The paper uses three complementary centrality measures (degree, betweenness, closeness), with each capturing different structural aspects. The ablation in Table 6 shows that while individual centralities help, combining all three consistently yields the best performance. For disconnected graphs, the authors properly handle the closeness centrality using harmonic closeness. This multi-faceted approach to identifying important nodes is more robust than relying on a single measure.
4. The centrality computation happens once per graph and can be cached, as stated in Section 3.5. While betweenness has O(nm) complexity, this is a one-time cost. During training, the only additional overhead is encoding κ additional subgraphs through the base GNN, which the authors claim is "roughly depth-invariant." This makes the approach practical for scenarios with repeated training on the same graphs, though the actual timing comparisons promised in Appendices C.4 and C.5 would be crucial for validation.

**Weaknesses:**

1. The theoretical analysis hinges on three assumptions (A1-A3) that could be justified better. A1 (task locality) assumes the label Y is independent of G\S* given S*, which fails for tasks requiring global graph properties like diameter or connectivity. A2 and A3 involve parameters γ and η that cannot be measured in practice. The authors claim their "ablations, perturbation tests, and depth-sensitivity studies collectively corroborate A1-A3," but the perturbation test (Table 5) only shows centrality masking hurts more than random masking. I am not sure this proves the label is independent of the rest of the graph.
2. The paper acknowledges in Section 4.5 that "deterministic tie-breaking by node index can, in principle, violate permutation invariance", a fundamental property for graph methods. While the authors suggest fixes, these are not implemented or evaluated.
3. Using centrality-based subgraph selection is well-established in graph analysis. I think the main contribution is applying this at the readout phase rather than during message passing, which is more of an engineering choice than a fundamental insight. The information-theoretic analysis (Theorem 1, Lemma 1) essentially states that adding more information doesn't decrease mutual information with the label, which is not very surprising.
4. While the paper claims improvements in 94/110 cases, many gains are marginal. In Table 1, for instance, several improvements are within one standard deviation (e.g., GIN on MUTAG: 81.0±10.2 vs 82.5±9.9). The paper admits that 16 configurations show no improvement or degradation but attributes this to "strict adherence to each model's original hyperparameters without any SP-specific tuning". Yet this same constraint should apply to the positive results too.
5. The paper fixes κ=2 and τ=4 across all experiments "for fair comparison" (Section 4.1), but this may be unfair since these are new hyperparameters that should be tuned per dataset like any other. The claim of using "original hyperparameters without any SP-specific tuning" is misleading when SP introduces its own hyperparameters. Additionally, comparing against methods like GCNII and U-Net that "exhibit poor performance" on ZINC/MolHIV (Table 4) inflates the relative performance of SP.
6. The O(nm) complexity for betweenness centrality becomes difficult for large graphs. While the authors mention "approximate centralities (e.g. PageRank proxies) are drop-in replacements" in Section 4.5, they provide no evaluation of how these approximations affect performance. For "web-scale graphs" mentioned in the conclusion, even the O(n^2) space for storing all-pairs shortest paths becomes infeasible.
7. The paper could include systematic studies on crucial design choices. There's no sensitivity analysis for κ and τ beyond the single ablation in Table 6. The fusion strategy is fixed to concatenation+MLP without exploring alternatives like attention-based fusion or learnable weighting. The choice of which centrality measures to use isn't justified beyond empirical performance. The enforced disjointness within centrality types (Algorithm 1) is not ablated to show its necessity.

**Questions:**

1. How does performance vary with different values of κ and τ? Why were these fixed at 2 and 4?
2. Can you provide statistical significance tests for the claimed improvements?
3. How does the method perform when task labels depend primarily on global graph properties rather than local substructures?
4. What is the actual computational overhead in practice for large-scale graphs (>10k nodes)?
5. How does the disjointness enforcement in Algorithm 1 affect the selected subgraphs when high-centrality nodes are clustered?
6. Have you considered learnable centrality measures rather than fixed graph-theoretic ones?

---

### Official Review · Reviewer_KhZ7 · 2025-11-01

**Soundness:** 3
**Presentation:** 3
**Contribution:** 3
**Rating:** 4
**Confidence:** 2

**Summary:**

This paper theoretically analyze the phenomenon of rank collapse problem in MPNNs, and we propose a lightweight plug-in called SP that allow GNNs to focus on subgraph informations.
Extensive experiments are conducted over 11 graph benchmarks with 13 GNN variants to validate the effectiveness of the proposed method.

**Strengths:**

1. The paper is well-written and the key ideas are clearly presented.
2. The authors provide theoretical analysis on the proposed method.
3. The experiments are comprehensive and the results are convincing.

**Weaknesses:**

Notations need to be carefully defined in the preliminaries and theorem sections. For example, I() is the mutual information?

The assumptions are too strong, especially A2. Besides the empirical evidence, is there any additional theoretical analysis to support or validate this assumption?

Moreover, theorem 1 seems intuitive, based on the assumption A2 and A3. What is the purpose to provide this theorem?

**Questions:**

See above.

---

### Meta-Review · Area_Chair_5ncz · 2025-12-04

**Summary:**

Given that three reviews rate the paper as “marginally below acceptance” and one is more clearly negative, the overall recommendation is rejection. The theoretical assumptions are quite strong and not operational in practice, the novelty over existing subgraph/pooling methods appears limited, and the empirical gains, while present, are modest and insufficiently analyzed. Moreover, the authors chose not to submit a rebuttal, leaving several key concerns unaddressed. I therefore recommend rejection at this stage.

**Reviewer Concerns:**

1. Theoretical Rigor & Assumption Validation: All reviewers highlighted that the core theoretical guarantees rely on unvalidated assumptions that cannot be measured or verified on real data.
2. Empirical Strength & Statistical Significance: Gains are frequently marginal (e.g., GIN on MUTAG: 81.0±10.2 vs. 82.5±9.9) and fall within standard deviations, with no statistical significance tests provided.
3. Novelty & Contextualization: The subgraph selection strategy is viewed as an engineering choice rather than a fundamental innovation, with limited differentiation from prior subgraph-based methods.

**Reviewer Scores:**

KhZ7: 4.
5agY: 4.
afsn: 4.
q2hC: 2.

---

### Decision · Program_Chairs · 2026-01-26

Reject